# COMMUNICATION EFFICIENT FEDERATED REPRESENTATION LEARNING

## ABSTRACT

The Federated Averaging (FedAvg) algorithm is a widely utilized technique in Federated Learning. It follows a recursive pattern where nodes perform a few local stochastic gradient descents (SGD), and then the central server updates the model by taking an average. The primary purpose of conducting model averaging is to mitigate the consensus error that arises between models across different nodes. In our empirical examination, it becomes evident that in non-iid data distribution setting, the consensus error in the initial layers of a deep neural network is considerably smaller than that observed in the final layers. This observation hints at the feasibility of applying a less intensive averaging approach for the initial layers. Typically, these layers are designed to extract meaningful representations from the neural network's input. To delve deeper into this phenomenon, we formally analyze it within the context of linear representation. We illustrate that increasing the number of local SGD iterations or reducing the frequency of averaging for the representation extractor leads to enhanced generalizability in the learned model produced by FedAvg's output. The paper is followed with experimental results showing the effectiveness of this method.

## 1 INTRODUCTION

Federated learning advocates that multiple clients collaboratively train machine learning models under the coordination of a parameter server (central aggregator) (McMahan et al., 2016). This approach has great potential for preserving the privacy of data stored at clients and reducing the communication cost of sending data from clients to a parameter server. Despite its promise, federated learning still suffers from high communication costs between clients and the parameter server.

In a federated learning setup, a parameter server oversees a global model and distributes it to participating clients. These clients then conduct local training using their own data. Then, the clients send their model updates to the parameter server, which aggregates them to a global model. This process continues until convergence. Exchanging machine learning models is costly especially for large models, which are typical in today's machine learning applications (Konecný et al., 2016; Zhang et al., 2013; Barnes et al., 2020; Braverman et al., 2015). Furthermore, the uplink bandwidth of clients may be limited, time-varying and expensive. Thus, there is an increasing interest in reducing the communication cost of federated learning especially by taking advantage of multiple local updates also known as "Local SGD" (Stich, 2018; Stich & Karimireddy, 2019; Wang & Joshi, 2018; Gholami & Seferoglu, 2023). The crucial questions in this context are (i) how long clients shall do Local SGD, and (ii) when they shall aggregate their local models. The goal of this paper is to address these questions to reduce communication costs without hurting performance.

The primary purpose of communication in federated learning is to periodically aggregate local models to reduce the consensus distance among clients. This practice helps maintain the overall optimization process on a trajectory toward global optimization. It is important to note that when the consensus distance among clients becomes substantial, the convergence rate reduces. This occurs as individual clients gradually veer towards their respective local optima without being synchronized with models from other clients. This issue is amplified when the data distribution among clients is non-iid. It has been demonstrated that the consensus distance is correlated to (i) the randomness in each client's own dataset, which causes variation in consecutive local gradients, as well as (ii) the dissimilarity in loss functions among clients due to non-iidness (Stich & Karimireddy, 2019; Gholami & Seferoglu, 2023). More specifically, the consensus distance at iteration $t$ is defined as

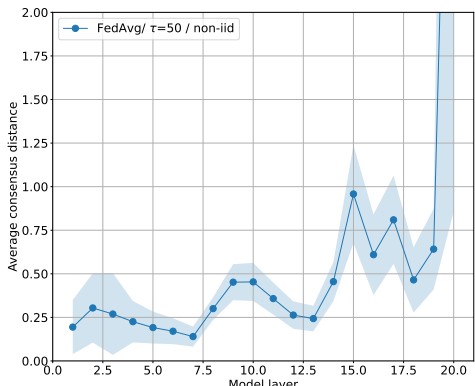

Figure 1: Average consensus distance over time for different layers measured while training a ResNet-20 by FedAvg on CIFAR-10 with 5 clients with non-iid data distribution over clients (2 classes per client). The early layers responsible for extracting representations exhibit lower levels of consensus distance.

$\frac{1}{K} \sum_{k=1}^{K} \|\hat{\boldsymbol{\theta}}_t - \boldsymbol{\theta}_{k,t}\|^2$, where $\hat{\boldsymbol{\theta}}_t = \frac{1}{K} \sum_{k=1}^{K} \boldsymbol{\theta}_{k,t}$, $K$ is the number of clients, and $\boldsymbol{\theta}_{k,t}$ is the local model at client $k$. Note that the consensus distance goes to zero when global aggregation is performed at each communication round. This makes the communication of models between clients and the parameter server crucial, but this introduces significant communication overhead.

Recent studies have demonstrated that the concept of representation learning is a promising approach to reduce the communication cost of federated learning (Collins et al., 2021). This is achieved by leveraging the shared representations that exist in the datasets of all clients. For example, let us consider a federated learning application for image classification, where different clients have datasets of different animals. Despite each client having a different dataset (one client has dog images, another has cat images, etc.), these images usually have common features such as an eye/ear shape. These shared features, typically extracted in the same way for different types of animals, require consistent layers of a neural network to extract them, whether the animal is a dog or a cat. As a result, these layers demonstrate similarity (i.e., less variation) across clients even when the datasets are non-iid. This implies that the consensus distance for this part of the model (feature extraction) is likely smaller. Based on these observations, Our key idea is to reduce the aggregation frequency of the layers that show high similarity, where these layers are updated locally between consecutive aggregations. This approach would reduce the communication cost of federated learning as some layers are aggregated, hence their parameters are exchanged, less frequently. This make it crucial to determine the layers that show high similarity. The next example scratches the surface of the problem for a toy example.

**Example 1.** *We consider a federated learning setup of five clients with a central parameter server to train a ResNet-20 (He et al., 2015) on a heterogeneous partition of CIFAR-10 dateset (Krizhevsky, 2009). We use Federated Averaging (FedAvg) (McMahan et al., 2016) as an aggregation algorithm as it stands as the dominant algorithm in federated learning. We applied FedAvg with 50 local steps prior to each averaging step, denoted as $\tau = 50$. Non-iidness is introduced by allocating 2 classes to each client. Finally, we evaluate the quantity of the average consensus distance for each model layer during the optimization, Fig. 1. It is clear that the initial layers have smaller consensus distance as compared to the final layers. This is due to initial layers' role in extracting representations from input data and their higher similarity across clients.*

The above example indicates that initial layers show higher similarity, so they can be aggregated less frequently. Additionally, several empirical studies (Reddi et al., 2021; Yu et al., 2020) show that FedAvg with multiple local updates per round learns a generalizable representation and is unexpectedly successful in non-iid federate learning. These studies encourage us to delve deeper into investigating how local updates and model aggregation frequency affect the model's representation extractor in terms of its generalization. In this paper, our objective is to provide analytical evidence demonstrating that increasing the number of local steps for the representation extractor leads to enhanced performance in the general setup, irrespective of the number of clients and in scenarios involving heterogeneous (non-iid) data distributions.

The contributions of this paper are in the following.

- We characterize the generalization error bound in federated learning utilizing a linear representation model under both iid and non-iid data distribution for the first time in the literature.

- We observe from our generalization error bound analysis that less frequent aggregations, hence more local updates (local SGDs), for the representation extractor (usually corresponds to initial layers) leads to the creation of more generalizable models particularly in non-iid scenarios (i.e., data distribution is heterogeneous).

- We design a novel FedALS (Federated Learning with Adaptive Local Steps) algorithm based on our generalization error bound analysis using representation learning. FedALS employs varying aggregation frequencies for different parts of the model.

- We evaluate the performance of FedALS using deep neural network model ResNet-20 He et al. (2015) for CIFAR-10, CIFAR-100 (Krizhevsky, 2009), and SVHN (Netzer et al., 2011) datasets. We consider both iid and non-iid data distributions. Experimental results confirm that the FedALS outperforms the baselines in non-iid setup.

## 2 RELATED WORK

There has been an increasing interest in distributed learning recently, largely driven by Federated Learning, which focuses on the analysis of decentralized algorithms utilizing Local SGD. Several studies have highlighted that these algorithms achieve convergence to a global optimum or a stationary point of the overall objective, particularly in convex or non-convex scenarios (Stich & Karimireddy, 2019; Stich, 2018; Gholami & Seferoglu, 2023; Lian et al., 2018; Kairouz et al., 2019). However, it is widely accepted that communication cost is the major bottleneck for this technique in large-scale optimization applications (Konecný et al., 2016; Lin et al., 2017). To tackle this issue, two primary strategies are put forth: the utilization of mini-batch parallel SGD, and the adoption of Local SGD. These approaches aim to enhance the equilibrium between computation and communication. Woodworth et al. (2020b;a) attempt to theoretically capture the distinction to comprehend under what circumstances Local SGD outperforms minibatch SGD.

Local SGD appears to be more intuitive compared to minibatch SGD, as it ensures progress towards the optimum even in cases where workers are not communicating and employing a mini-batch size that is too large may lead to a decrease in performance (Lin et al., 2017). However, due to the fact that individual gradients for each worker are computed at distinct instances, this technique brings about residual errors. As a result, a compromise arises between reducing communication rounds and introducing supplementary errors into the gradient estimations. This becomes increasingly significant when data is unevenly distributed across nodes. There are several decentralized algorithms that have been shown to mitigate heterogeneity (Karimireddy et al., 2019; Liu et al., 2023) . One prominent example is the Stochastic Controlled Averaging algorithm (SCAFFOLD) (Karimireddy et al., 2019), which addresses the node drift caused by non-iid characteristics of data distribution. They establish the notion that SCAFFOLD demonstrates a convergence rate at least equivalent to that of SGD, ensuring convergence even when dealing with highly non-iid datasets.

However, despite these factors, multiple empirical investigations (Reddi et al., 2021; Yu et al., 2020) have noted that the model trained using FedAvg and incorporating multiple Local SGD per round exhibits unexpected effectiveness when subsequently fine-tuned for individual clients in non-iid FL setting. This implies that the utilization of FedAvg with several local updates proves effective in acquiring a valuable data representation, which can later be employed on each node for downstream tasks. Following this line of reasoning, our justification will be based on the argument that the Local SGD component of FedAvg contributes to improving performance in heterogeneous scenarios by facilitating the acquisition of models with enhanced generalizability.

An essential characteristic of machine learning systems is their capacity to extend their performance to novel and unseen data. While these systems are trained on a specific dataset, they are expected to perform effectively on new data points that were not included during training. This capacity, referred to as generalization, can be expressed within the framework of statistical learning theory by assessing the algorithm's generalization error. There has been a line of research to characterize generalization bound in FL. Barnes et al. (2022a) considered this problem and gave upper bounds on the expected generalization error for FL in iid setting, which demonstrates an improved dependence of $\frac{1}{K}$ on the number of nodes. Motivated by this work, we build our results for both iid and non-iid settings for the linear representation model class.

## 3 Background and Problem Statement

### 3.1 Preliminaries and Notation

We consider that we have $K$ clients/nodes in our system, and each node has its own portion of the dataset. For example, node $k$ has a local dataset $\boldsymbol{S}_k = \{\boldsymbol{z}_{k,1}, ..., \boldsymbol{z}_{k,n_k}\}$, where $\boldsymbol{z}_{k,i} = (\boldsymbol{x}_{k,i}, \boldsymbol{y}_{k,i})$ is drawn from a distribution $\mathcal{D}_k$ over $\mathcal{X} \times \mathcal{Y}$, where $\mathcal{X}$ is the input space and $\mathcal{Y}$ is the label space. We consider $\mathcal{X} \subseteq \mathbb{R}^d$ and $\mathcal{Y} \subseteq \mathbb{R}$. The size of the local dataset at node $k$ is $n_k$. The dataset across all nodes is defined as $\boldsymbol{S} = \{\boldsymbol{S}_1, ..., \boldsymbol{S}_K\}$. Data distribution across the nodes could be independent and identically distributed (iid) or non-iid. In iid setting, we assume that $\mathcal{D}_1 = ... = \mathcal{D}_K = \mathcal{D}$ holds. On the other hand, non-iid setting covers all possible distributions and cases, where $\mathcal{D}_1 = ... = \mathcal{D}_K = \mathcal{D}$ does not hold.

We assume that $M_{\boldsymbol{\theta}} = \mathcal{A}(\boldsymbol{S})$ represents the output of a possibly stochastic function denoted as $\mathcal{A}(\boldsymbol{S})$, where $M_{\boldsymbol{\theta}} : \mathcal{X} \to \mathcal{Y}$ represents the learned model parameterized by $\boldsymbol{\theta}$. We consider a real-valued loss function denoted as $l(M_{\boldsymbol{\theta}}, \boldsymbol{z})$, which assesses the model $M_{\boldsymbol{\theta}}$ based on a sample $\boldsymbol{z}$.

### 3.2 Generalization Error

We first define an empirical risk on dataset $\boldsymbol{S}$ as

$$R_{\boldsymbol{S}}(M_{\boldsymbol{\theta}}) = \mathbb{E}_{k \sim \mathcal{K}} R_{\boldsymbol{S}_k}(M_{\boldsymbol{\theta}}) = \mathbb{E}_{k \sim \mathcal{K}} \frac{1}{n_k} \sum_{i=1}^{n_k} l(M_{\boldsymbol{\theta}}, \boldsymbol{z}_{k,i}), \tag{1}$$

where $\mathcal{K}$ is an arbitrary distribution on nodes, and $R_{\boldsymbol{S}_k}(M_{\boldsymbol{\theta}})$ is the empirical risk for model $M_{\boldsymbol{\theta}}$ on local dataset $\boldsymbol{S}_k$. We further define a population risk for model $M_{\boldsymbol{\theta}}$ as

$$R(M_{\boldsymbol{\theta}}) = \mathbb{E}_{k \sim \mathcal{K}} R_k(M_{\boldsymbol{\theta}}) = \mathbb{E}_{k \sim \mathcal{K}, \boldsymbol{z} \sim \mathcal{D}_k} l(M_{\boldsymbol{\theta}}, \boldsymbol{z}), \tag{2}$$

where $R_k(M_{\boldsymbol{\theta}})$ is the population risk on node $k$'s data distribution.

Now, we can define the generalization error for dataset $\boldsymbol{S}$ and function $\mathcal{A}(\boldsymbol{S})$ as

$$\Delta_{\mathcal{A}}(\boldsymbol{S}) = R(\mathcal{A}(\boldsymbol{S})) - R_{\boldsymbol{S}}(\mathcal{A}(\boldsymbol{S})). \tag{3}$$

The expected generalization error is expressed as

$$\mathbb{E}_{\{\boldsymbol{S}_k \sim \mathcal{D}_k^{n_k}\}_{k=1}^K} \Delta_{\mathcal{A}}(\boldsymbol{S}) = \mathbb{E}_{\{\boldsymbol{S}_k \sim \mathcal{D}_k^{n_k}\}_{k=1}^K} \left[ R(\mathcal{A}(\boldsymbol{S})) - R_{\boldsymbol{S}}(\mathcal{A}(\boldsymbol{S})) \right]. \tag{4}$$

If a model has a small generalization error, it is considered that the model enjoys effective generalization. Existing approaches for training models usually minimize the empirical risk or its variants. Consequently, a small empirical risk and a reduced generalization error correspond to a low population risk. Small population risk shows how good the model is in the test phase as it denotes the loss that occurs when new samples are randomly drawn from the distribution. Thus, there is an increasing interest in establishing an upper limit for the generalization error and understanding the underlying factors that affect the generalization error. The generalization error analysis is also important to quantitatively assess the generalization characteristics of trained models, provide reliable guarantees concerning their anticipated performance quality, and design new models and systems.

Our goal in this paper is to *use generalization error bounds to design communication efficient federated learning when data is distributed heterogeneously over nodes,* i.e., *non-iid setup.*

### 3.3 Federated Learning

We consider a federated learning scenario with $K$ nodes/clients and a centralized parameter server. The nodes update their localized models to minimize their empirical risk $R_{\boldsymbol{S}_k}(M_{\boldsymbol{\theta}})$ on local dataset $\boldsymbol{S}_k$, while the parameter server aggregates the local models to minimize the empirical risk $R_{\boldsymbol{S}}(M_{\boldsymbol{\theta}})$. Due to connectivity and privacy constraints, the clients do not exchange their data with each other. One of the most widely used federated learning algorithms is FedAvg (McMahan et al., 2016), which we explain in detail next.

At round $r$ of FedAvg, each node $k$ trains its model $M_{\boldsymbol{\theta}_{k,r}} = \mathcal{A}_{k,r}(\boldsymbol{S}_k)$ locally using the function/algorithm $\mathcal{A}_{k,r}$. The local models $M_{\boldsymbol{\theta}_{k,r}}$ are transmitted to the central parameter server, which merges the received local models to aggregated model parameters $\hat{\boldsymbol{\theta}}_{r+1} = \hat{\mathcal{A}}(\boldsymbol{\theta}_{1,r}, ..., \boldsymbol{\theta}_{K,r})$, where

$\hat{\mathcal{A}}$ is the aggregation function. In FedAvg, the aggregation function calculates an average, so the aggregated model is expressed as

$$\hat{\boldsymbol{\theta}}_{r+1} = \frac{1}{K} \sum_{k=1}^{K} \boldsymbol{\theta}_{k,r}. \tag{5}$$

Subsequently, the aggregated model is transmitted to all nodes. This process continues for $R$ rounds. The final model after $R$ rounds of FedAvg is $\mathcal{A}(S)$.

The local models are usually trained using stochastic gradient descent (SGD) at each node. To reduce the communication cost needed between the nodes and the parameter server, each node executes multiple SGD steps using its local data after receiving an aggregated model from the parameter server. To be precise, we define the aggregated model parameters at round $r$ as $\hat{\boldsymbol{\theta}}_r$. Specifically, upon receiving $\hat{\boldsymbol{\theta}}_r$, node $k$ computes

$$\boldsymbol{\theta}_{k,r,t+1} = \boldsymbol{\theta}_{k,r,t} - \frac{\eta}{|\mathcal{B}_{k,r,t}|} \sum_{i \in \mathcal{B}_{k,r,t}} \nabla l(M_{\boldsymbol{\theta}_{k,r,t}}, \boldsymbol{z}_{k,i}) \tag{6}$$

for $t = 0, \ldots, \tau - 1$, where $\tau$ is the number of local SGD steps, $\boldsymbol{\theta}_{k,r,0}$ is defined as $\boldsymbol{\theta}_{k,r,0} = \hat{\boldsymbol{\theta}}_r$, $\eta$ is the learning rate, $\mathcal{B}_{k,r,t}$ is the batch of samples used in local step $t$ of round $r$ in node $k$, $\nabla$ is the gradient, and $|.|$ shows the size of a set. Upon completing the local steps in round $r$, each node transmits $\boldsymbol{\theta}_{k,r} = \boldsymbol{\theta}_{k,r,\tau}$ to the parameter server to calculate $\hat{\boldsymbol{\theta}}_{r+1}$ as in (5).

## 3.4 REPRESENTATION LEARNING

Our approach for analyzing the generalization error bounds for federated learning, by specifically focusing on FedAvg, uses representation learning, which we explain next.

We consider a class of models that consist of a representation extractor (*e.g.,* ResNet). Let $\boldsymbol{\theta}$ be the model $M_{\boldsymbol{\theta}}$'s parameters. We can decompose $\boldsymbol{\theta}$ into two sets: $\boldsymbol{\phi}$ containing the representation extractor's parameters and $\boldsymbol{h}$ containing the head parameters, *i.e.,* $\boldsymbol{\theta} = [\boldsymbol{\phi}, \boldsymbol{h}]$. $M_{\boldsymbol{\phi}}$ is a function that maps from the original input space to a low-dimensional representation space, *i.e.,* $M_{\boldsymbol{\phi}} : \mathbb{R}^d \to \mathbb{R}^{d'}$, $d' \ll d$. The function $M_{\boldsymbol{h}}$ performs a mapping from the representation space to the label space, which can be expressed as $M_{\boldsymbol{h}} : \mathbb{R}^{d'} \to \mathbb{R}$.

For any $\boldsymbol{x} \in \mathcal{X}$, the output of the model is $M_{\boldsymbol{\theta}}(\boldsymbol{x}) = (M_{\boldsymbol{\phi}} \circ M_{\boldsymbol{h}})(\boldsymbol{x}) = M_{\boldsymbol{h}}(M_{\boldsymbol{\phi}}(\boldsymbol{x}))$. For instance, if $M_{\boldsymbol{\theta}}$ is a neural network, $M_{\boldsymbol{\phi}}$ represents several initial layers of the network, which are typically designed to extract meaningful representations from the neural network's input. On the other hand, $M_{\boldsymbol{h}}$ denotes the final few layers that lead to the network's output.

## 4 GENERALIZATION BOUND ANALYSIS FOR FEDAVG VIA REPRESENTATION LEARNING

In this section, we derive generalization bounds for FedAvg by using representation learning for the first time in the literature. First, we start with one-round FedAvg and analyze its generalization bound. Then, we extend our analysis to $R-$round FedAvg.

### 4.1 ONE-ROUND GENERALIZATION BOUND

We consider the standard algorithmic representation learning analysis, where the representation is a linear map from the original input space to a low-dimensional space (Du et al., 2020; Collins et al., 2021; 2022). Namely, the representation function class is defined as $M_{\boldsymbol{\phi}} = \{\boldsymbol{x} \to \boldsymbol{x}B \mid B \in \mathbb{R}^{d \times d'}\}$. This is followed by a linear head, *i.e.,* $M_{\boldsymbol{h}} = \{\boldsymbol{x} \to \boldsymbol{x}\boldsymbol{w} \mid \boldsymbol{w} \in \mathbb{R}^{d'}\}$.

In the following lemma, we determine the generalization bound for one round of FedAvg in both iid and non-iid settings.

**Lemma 4.1.** *Consider that $M_{\boldsymbol{\theta}}(\boldsymbol{x}) = \boldsymbol{x}B\boldsymbol{w}$ is the model class, $l(M_{\boldsymbol{\theta}}, \boldsymbol{z}) = (\boldsymbol{y} - \boldsymbol{x}B\boldsymbol{w})^2$ is the loss function, $M_{\boldsymbol{\theta}_k} = \mathcal{A}_k(\boldsymbol{S}_k)$ represents the model obtained from Empirical Risk Minimization (ERM) algorithm on local dataset $\boldsymbol{S}_k$, i.e., $M_{\boldsymbol{\theta}_k} = \arg\min_M \sum_{i=1}^{n_k} l(M, \boldsymbol{z}_{k,i})$, $M_{\hat{\boldsymbol{\theta}}} = \mathcal{A}(\boldsymbol{S})$ is the model after one round of FedAvg ($\hat{\boldsymbol{\theta}} = \mathbb{E}_{k \sim \mathcal{K}} \boldsymbol{\theta}_k$), and $\mathcal{K}$ is a uniform distribution overs all nodes.*

*Then, for iid data distribution over nodes, the expected generalization error is*

$$\mathbb{E}_{\{\boldsymbol{S}_k \sim \mathcal{D}_k^{n_k}\}_{k=1}^{K}} \Delta_{\mathcal{A}}(\boldsymbol{S}) \leq \frac{1}{K^2} \sum_{k=1}^{K} \mathbb{E}_{\boldsymbol{S}_k \sim \mathcal{D}_k^{n_k}} \Delta_{\mathcal{A}_k}(\boldsymbol{S}_k), \tag{7}$$

*and the expected generalization error in non-iid case is*

$$\mathbb{E}_{\{\boldsymbol{S}_k \sim \mathcal{D}_k^{n_k}\}_{k=1}^{K}} \Delta_{\mathcal{A}}(\boldsymbol{S}) \leq \frac{1}{K} \sum_{k=1}^{K} \mathbb{E}_{\boldsymbol{S}_k \sim \mathcal{D}_k^{n_k}} \Delta_{\mathcal{A}_k}(\boldsymbol{S}_k). \tag{8}$$

Lemma 4.1 shows that the limit of the expected generalization error bound decreases as the number of nodes $K$ increases in iid case in (7). As a result, after each averaging process carried out by the central parameter server, the generalization error is reduced by a factor of $K$ in iid case. On the other hand, we do not see a similar behavior in non-iid case in (8). In other words, the expected generalization error bound does not necessarily decrease if the number of nodes increases. These results show why FedAvg works well in iid setup, but not necessarily in non-iid setup. This observation motivates us to design a new federated learning approach for non-iid setup. The question in this context is what should be the new federated learning design. To answer this question, we analyze $R-$round generalization bound in the next section.

## 4.2 $R-$ROUND GENERALIZATION BOUND

Now we turn our attention to a more complicated scenario; $R$-round FedAvg. A similar $R$-round generalization bound analysis is considered in Barnes et al. (2022a), but without representation learning, which is crucial in our proposed federated learning mechanism in Section 5.

In this setup, after $R$ rounds, there is a sequence of weights $\{\hat{\boldsymbol{\theta}}_r\}_{r=1}^{R}$ and the final model is $\hat{\boldsymbol{\theta}}_R$. We consider that at round $r$, each node constructs its updated model as in (6) by taking $\tau$ gradient steps starting from $\hat{\boldsymbol{\theta}}_r$ with respect to $\tau$ random mini-batches $Z_{k,r} = \bigcup\{\mathcal{B}_{k,r,t}\}_{t=0}^{\tau-1}$ drawn from the local dataset $\boldsymbol{S}_k$. For this type of iterative algorithm, we consider the following averaged empirical risk

$$\frac{1}{KR} \sum_{r=1}^{R} \sum_{k=1}^{K} \frac{1}{|Z_{k,r}|} \sum_{i \in Z_{k,r}} l(M_{\hat{\boldsymbol{\theta}}_r}, \boldsymbol{z}_{k,i}). \tag{9}$$

The corresponding generalization error is

$$\Delta_{FedAvg}(\boldsymbol{S}) = \frac{1}{KR} \sum_{r=1}^{R} \sum_{k=1}^{K} \left( \mathbb{E}_{\boldsymbol{z} \sim \mathcal{D}_k} l(M_{\hat{\boldsymbol{\theta}}_r}, \boldsymbol{z}) - \frac{1}{|Z_{k,r}|} \sum_{i \in Z_{k,r}} l(M_{\hat{\boldsymbol{\theta}}_r}, \boldsymbol{z}_{k,i}) \right). \tag{10}$$

Note that the expression in (10) differs slightly from the end-to-end generalization error that would be obtained by considering the final model $M_{\hat{\boldsymbol{\theta}}_R}$ and the entire dataset $\boldsymbol{S}$. More specifically, (10) is an average of the generalization errors measured at each round. We anticipate that the generalization error diminishes with the increasing number of data samples, so this generalization error definition yields to a more cautious upper limit and serves as a sensible measure. The next theorem characterizes the expected generalization error bounds for $R-$Round FedAvg in iid and non-iid settings.

**Theorem 4.2.** *Consider that $M_{\boldsymbol{\theta}}(\boldsymbol{x}) = \boldsymbol{x}B\boldsymbol{w}$ is the model class, $l(M_{\boldsymbol{\theta}}, \boldsymbol{z}) = (\boldsymbol{y} - \boldsymbol{x}B\boldsymbol{w})^2$ is the loss function, and $\mathcal{K}$ is a uniform distribution over all nodes. Local models at round $r$ are calculated by doing multiple local steps as in (6). The aggregated model at round $r$ is $M_{\hat{\boldsymbol{\theta}}_r}$ is obtained by performing FedAvg in (5) where the data points used in round $r$ (i.e., $Z_{k,r}$) are sampled without replacement.*

*For iid data distribution over nodes, the average generalization error bound is expressed as*

$$\mathbb{E}_{\{\boldsymbol{S}_k \sim \mathcal{D}_k^{n_k}\}_{k=1}^{K}} \Delta_{FedAvg}(\boldsymbol{S}) \leq \frac{1}{R} \sum_{r=1}^{R} \frac{1}{K^2} \sum_{k=1}^{K} \sqrt{\frac{\mathcal{C}(M_{\boldsymbol{\theta}})}{|Z_{k,r}|}}, \tag{11}$$

*while in non-iid case, the average generalization error bound is expressed as*

$$\mathbb{E}_{\{\boldsymbol{S}_k \sim \mathcal{D}_k^{n_k}\}_{k=1}^{K}} \Delta_{FedAvg}(\boldsymbol{S}) \leq \frac{1}{R} \sum_{r=1}^{R} \frac{1}{K} \sum_{k=1}^{K} \sqrt{\frac{\mathcal{C}(M_{\boldsymbol{\theta}})}{|Z_{k,r}|}}, \tag{12}$$

*where $\mathcal{C}(M_{\boldsymbol{\theta}})$ shows the complexity of the model class of $M_{\boldsymbol{\theta}}$.*

---

**Algorithm 1** FedALS

---

**Input**: Initial model $\{\boldsymbol{\theta}_{k,1,0} = [\boldsymbol{\phi}_{k,1,0}, \boldsymbol{h}_{k,1,0}]\}_{k=1}^{K}$, Learning rate $\eta$, number of local steps for the head model $\tau$, adaptation coefficient $\alpha$.

  1: **for** Round $r$ in $1, ..., R$ **do**
  2:     **for** Node $k$ in $1, ..., K$ **in parallel do**
  3:         **for** Local step $t$ in $0, ..., \tau - 1$ **do**
  4:             Sample the batch $\mathcal{B}_{k,r,t}$ from $\mathcal{D}_k$.
  5:             $\boldsymbol{\theta}_{k,r,t+1} = \boldsymbol{\theta}_{k,r,t} - \frac{\eta}{|\mathcal{B}_{k,r,t}|} \sum_{i \in \mathcal{B}_{k,r,t}} \nabla l(M_{\boldsymbol{\theta}_{k,r,t}}, \boldsymbol{z}_{k,i})$
  6:             **if** $\mod(r\tau + t, \tau) = 0$ **then**
  7:                 $\boldsymbol{h}_{k,r,t} \leftarrow \frac{1}{K} \sum_{k=1}^{K} \boldsymbol{h}_{k,r,t}$
  8:             **else if** $\mod(r\tau + t, \alpha\tau) = 0$ **then**
  9:                 $\boldsymbol{\phi}_{k,r,t} \leftarrow \frac{1}{K} \sum_{k=1}^{K} \boldsymbol{\phi}_{k,r,t}$
 10:     $\boldsymbol{\theta}_{k,r+1,0} = \boldsymbol{\theta}_{k,r,\tau}$
 11: **return** $\hat{\boldsymbol{\theta}}_R = \frac{1}{K} \sum_{k=1}^{K} \boldsymbol{\theta}_{k,R,\tau}$

---

The generalization error bounds in (11), and (12) depend on the following parameters: (i) number of rounds; $R$, (ii) number of samples used in every round; $|Z_{k,r}|$, and (iii) the complexity of the model class; $\mathcal{C}(M_{\boldsymbol{\theta}})$. We note that (11), and (12) also depend on $K$, but this dependence is similar to the discussion we had for one-round generalization, so we skip it here.

The complexity of the model class and the number of samples used in every round are crucial to minimize the generalization error bound especially in non-iid case in (12) noting that the generalization error bound has a reduction factor of $\frac{1}{K}$ in iid setup. Some common complexity measures in the literature include the number of parameters (classical VC Dimension (Shalev-Shwartz & Ben-David, 2014)), parameter norms (*e.g.*, $l_1, l_2$, spectral) (Bartlett, 1997), or other potential complexity measures (Lipschitzness, Sharpness, . . . ) (Neyshabur et al., 2017; Dziugaite & Roy, 2017; Nagarajan & Kolter, 2019; Wei & Ma, 2019; Norton & Royset, 2019; Foret et al., 2021).

Independent from a specific complexity measure, a model in representation learning can be divided into two parts: (i) $M_{\boldsymbol{\phi}}$, which is the representation extractor, and (ii) $M_{\boldsymbol{h}}$, which maps the representation to an output. The complexities of these parts follow $\mathcal{C}(M_{\boldsymbol{h}}) \ll \mathcal{C}(M_{\boldsymbol{\phi}})$ as $d' \ll d$.

Our key intuition in this paper is that we can *reduce the aggregation frequency of $M_{\boldsymbol{\phi}}$, which leads to a larger $\tau$ and $|Z_{k,r}|$, hence smaller generalization error bound according to (12)*.[1] We note that the aggregation frequency of $M_{\boldsymbol{\phi}}$ can not be reduced arbitrarily as it would increase the empirical risk. As seen, there is a nice trade-off between aggregation frequency of $M_{\boldsymbol{\phi}}$ and population risk. In the next section, we design our Federated Learning with Adaptive Local Steps (FedALS) algorithm by taking into account this trade-off.

## 5 FEDALS: FEDERATED LEARNING WITH ADAPTIVE LOCAL STEPS

Theorem 4.2 and our key intuition above demonstrate that more local SGD steps (less aggregations at the parameter server) are necessary for representation extractor $M_{\boldsymbol{\phi}}$ as compared to the model's head $M_{\boldsymbol{h}}$ to reduce generalization error bound. This approach, since it will reduce the aggregation frequency of $M_{\boldsymbol{\phi}}$, will also reduce the communication cost of federated learning. In this context, the crucial question is to determine how frequently $M_{\boldsymbol{\phi}}$ and $M_{\boldsymbol{h}}$ should be communicated and aggregated to keep the generalization error bound small. Our FedALS design provides a solution to this question.

The main idea of FedALS is to maintain a uniform generalization error across both components ($M_{\boldsymbol{\phi}}$ and $M_{\boldsymbol{h}}$) of the model. This can be achieved if $\tau_{M_{\boldsymbol{\phi}}}$ is set on the order of $\frac{\mathcal{C}(M_{\boldsymbol{\phi}})}{\mathcal{C}(M_{\boldsymbol{h}})}\tau_{M_{\boldsymbol{h}}}$, where $\tau_M$ denotes the number of local iterations in a single round for the model $M$ while $\tau_{M_{\boldsymbol{\phi}}}$ and $\tau_{M_{\boldsymbol{h}}}$ are the corresponding number of local iterations for $M_{\boldsymbol{\phi}}$ and $M_{\boldsymbol{h}}$, respectively. Following this approach, we designed FedALS in Algorithm 1.

---

[1]We do not reduce the aggregation frequency of $M_{\boldsymbol{h}}$ as its complexity, so its contribution to generalization error, is small.

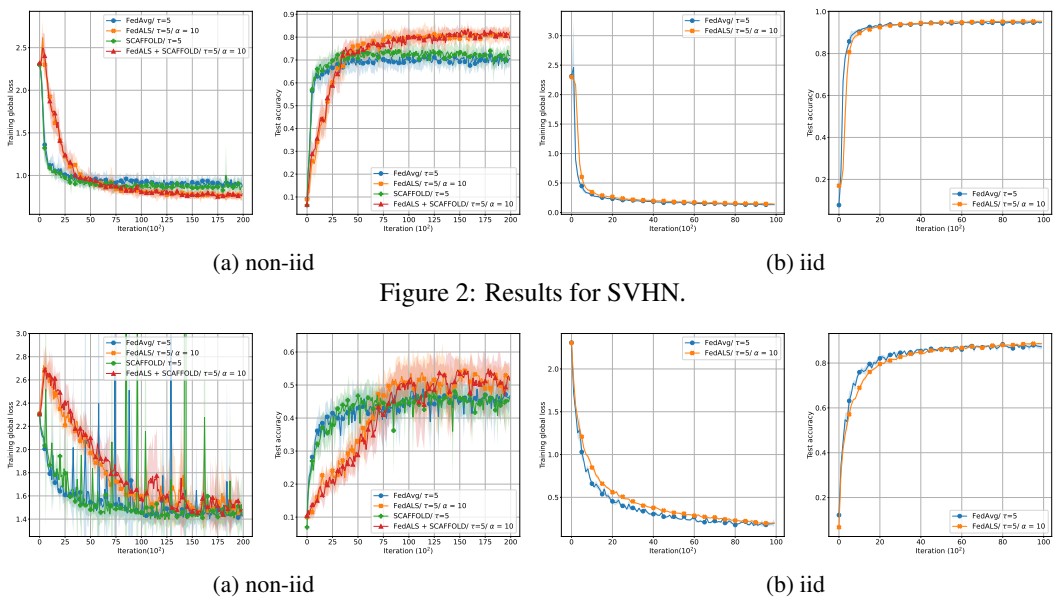

(a) non-iid            (b) iid

Figure 2: Results for SVHN.

(a) non-iid            (b) iid

Figure 3: Results for CIFAR-10.

FedALS in Algorithm 1 divides the model into two parts: 1- the representation extractor, denoted as $M_{\phi}$, and 2- the head, denoted as $M_{h}$. Additionally, we introduce the parameter $\alpha = \frac{\tau_{M_{\phi}}}{\tau_{M_{h}}}$ as an adaptation coefficient, which can be regarded as a hyperparameter for estimating $\frac{\mathcal{C}(M_{\phi})}{\mathcal{C}(M_{h})}$, considering that determining this value is not straightforward.

## 6 EXPERIMENTAL RESULTS

In this section, we evaluate the performance of FedALS by employing ResNet-20 as a deep neural network architecture. We regard the convolutional layers as the representation extractor and the final dense layers as the head of the model. The datasets we used are CIFAR-10, CIFAR-100 and SVHN. The experimentation was conducted on a network comprising five nodes alongside a central server. A batch size of $64$ per node was configured. To grid-search the learning rate, we try each experiment with multiplying and dividing the learning rate by powers of two. Additionally, the momentum was set to $0.9$, and the weight decay to $10^{-4}$.

The experiments are conducted on Ubuntu $20.04$ using 36 Intel Core i9-10980XE processors and 5 GeForce RTX 2080 graphics cards. We repeat each experiment 20 times and present the error bars associated with the randomness of the optimization. In every figure, we include the average and standard deviation error bars.

### 6.1 FEDALS IN NON-IID SETTING

In this section, we allocate the dataset to nodes using a non-iid approach. This is achieved by initially sorting the data based on their labels and subsequently dividing it among nodes following this sorted sequence.

In this scenario, we can observe in Fig. 2a, 3a the anticipated performance improvement through the incorporation of different local steps across the model. By utilizing parameters $\tau = 5$ and $\alpha = 10$ in FedALS, it becomes apparent that aggregation and communication costs are reduced as compared to FedAvg with the same $\tau$ value of 5. This implies that the initial layers perform aggregation at every 50 iterations. This reduction in the number of communications is accompanied by enhanced model generalization stemming from the larger number of local steps in the initial layers, which contributes to an overall performance enhancement. Thus, our approach in FedALS is beneficial for both communication efficiency and enhancing model generalization performance simultaneously.

### 6.2 FEDALS IN IID SETTING

The results for the iid setting are presented in Fig. 2b, 3b. In order to obtain these results, the data is shuffled, and then evenly divided among nodes. We note that in this situation, the performance improvement of FedALS is negligible. We expected this result as the generalization error bound (7)

Table 1: Test accuracy for 5 nodes FL, after training ResNet-20 for $10^4$ iterations in iid and $2 \times 10^4$ iterations in non-iid setting with $\tau = 5$ and $\alpha = 10$.

| dataset | FedAvg | | FedALS | | SCAFFOLD | FedALS + SCAFFOLD |
| --- | --- | --- | --- | --- | --- | --- |
| | iid | non-iid | iid | non-iid | non-iid | non-iid |
| SVHN | 94.74 | 70.10 | **95.38** | 80.97 | 71.79 | **81.07** |
| CIFAR-10 | 87.62 | 48.38 | **88.65** | **52.51** | 46.51 | 51.89 |
| CIFAR-100 | 59.89 | 41.68 | **61.20** | 48.04 | 41.98 | **48.20** |

decreases after each aggregation on the order of $K$. Thus, the improvement of the generalization error using FedALS approach is negligible in this setup.

### 6.3 COMPARED TO SCAFFOLD

Karimireddy et al. (2019) introduced an innovative technique called SCAFFOLD, which employs some control variables for variance reduction to address the issue of "client-drift" in local updates. This drift happens when data is heterogeneous (non-iid), causing individual nodes/clients to converge towards their local optima rather than the global optima. While this approach is a significant theoretical advancement in achieving independence from loss function disparities among nodes, it hinges on the assumption of smoothness in the loss functions, which might not hold true for practical deep learning problems in the real world. Additionally, since SCAFFOLD requires the transmission of control variables to the central server, which is of the same size as the models themselves, it results in approximately twice the communication overhead when compared to FedAvg.

Let us consider Fig. 2a, 3a to notice that in real-world deep learning situations, FedALS enhances performance significantly, while SCAFFOLD exhibits slight improvements in specific scenarios. Moreover, we integrated FedALS and SCAFFOLD to concurrently leverage both approaches. The results of the test accuracy in different cases are summarized in Table 1.

### 6.4 THE ROLE OF $\alpha$

As shown in Fig. 4, and Table 2, it becomes evident that when we increase $\alpha$ from 1 (FedAvg), we initially witness an enhancement in accuracy owing to improved generalization. However, beyond a certain threshold ($\alpha = 10$), further increment in $\alpha$ ceases to contribute to performance improvement. This is due to the adverse impact of a high number of local steps on optimization performance. The trade-off we discussed in the earlier sections is evident in this context.

## 7 CONCLUSION

In this research, we examined how Local SGD affects the generalization aspects of federated learning within the context of representation learning. Our findings demonstrate that achieving a stronger generalization bound in this system necessitates a greater number of local steps for the representation extractor component of the model compared to the model's final layers. This insight led us to develop the FedALS algorithm, which centers around the concept of increasing local steps for the initial layers of the deep learning model while conducting more averaging for the final layers.

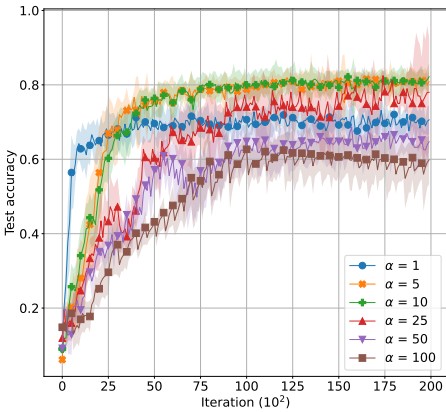

Figure 4: The role of $\alpha$ in training ResNet-20 for SVHN non-iid with $\tau = 5$.

| Value of $\alpha$ | Accuracy |
| --- | --- |
| 1 | 70.10 |
| 5 | 79.92 |
| 10 | **80.97** |
| 25 | 76.80 |
| 50 | 63.77 |
| 100 | 58.37 |

Table 2: The accuracy achieved after training ResNet-20 for $2 \times 10^4$ iterations on SVHN non-iid with $\tau = 5$ and variable $\alpha$.

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

# A   PROOF OF LEMMA 4.1

We first state and prove the following Lemma that will be used in the proof of lemma 4.1.

**Lemma A.1** (Leave-one-out). *[Expansion of theorem 1 in Barnes et al. (2022b)]*

*Let $S'_k = (z'_{k,1}, ..., z'_{k,n_k})$, where $z'_{k,i}$ is sampled from $\mathcal{D}_k$. Let also define $S' = (S'_1, ..., S'_K)$. Denote $S_k^{(i)} = (z_{k,1}, ..., z'_{k,i}, ..., z_{k,n_k})$ to represent the alteration of $S_k$ wherein the $i$-th data sample is replaced with $z'_{k,i}$. Also define $S^{(k,i)} = (S_1, ..., S_k^{(i)}, ..., S_K)$. Then*

$$\mathbb{E}_{\{S_k \sim \mathcal{D}_k^{n_k}\}_{k=1}^K} \Delta_{\mathcal{A}}(S) = \mathbb{E}_{k \sim \mathcal{K}, \{S_k, S'_k \sim \mathcal{D}_k^{n_k}\}_{k=1}^K} \left[ \frac{1}{n_k} \sum_{i=1}^{n_k} \left( l(\mathcal{A}(S), z'_{k,i}) - l(\mathcal{A}(S^{(k,i)}), z'_{k,i}) \right) \right]. \tag{13}$$

*Proof.* We have

$$\mathbb{E}_{\{S_k \sim \mathcal{D}_k^{n_k}\}_{k=1}^K} R(\mathcal{A}(S)) = \mathbb{E}_{k \sim \mathcal{K}, \{S_k, S'_k \sim \mathcal{D}_k^{n_k}\}_{k=1}^K} l(\mathcal{A}(S), z'_{k,i}). \tag{14}$$

Also, observe that

$$\mathbb{E}_{\{S_k \sim \mathcal{D}_k^{n_k}\}_{k=1}^K} R_S(\mathcal{A}(S)) = \mathbb{E}_{k \sim \mathcal{K}, \{S_k \sim \mathcal{D}_k^{n_k}\}_{k=1}^K} \left[ \frac{1}{n_k} \sum_{i=1}^{n_k} l(\mathcal{A}(S), z_{k,i}) \right] \tag{15}$$

$$= \mathbb{E}_{k \sim \mathcal{K}, \{S_k, S'_k \sim \mathcal{D}_k^{n_k}\}_{k=1}^K} \left[ \frac{1}{n_k} \sum_{i=1}^{n_k} l(\mathcal{A}(S^{(k,i)}), z'_{k,i}) \right]. \tag{16}$$

Putting 14, and 16 together, by the definition of the expected generalization error we get

$$\mathbb{E}_{\{S_k \sim \mathcal{D}_k^{n_k}\}_{k=1}^K} \Delta_{\mathcal{A}}(S) = \mathbb{E}_{k \sim \mathcal{K}, \{S_k, S'_k \sim \mathcal{D}_k^{n_k}\}_{k=1}^K} \left[ \frac{1}{n_k} \sum_{i=1}^{n_k} \left( l(\mathcal{A}(S), z'_{k,i}) - l(\mathcal{A}(S^{(k,i)}), z'_{k,i}) \right) \right]. \tag{17}$$

$\square$

In the following lemma, we determine the generalization bound for one round of FedAvg in both iid and non-iid settings (Lemma 4.1).

**Lemma A.2.** *Consider that $M_{\boldsymbol{\theta}}(\boldsymbol{x}) = \boldsymbol{x}B\boldsymbol{w}$ is the model class, $l(M_{\boldsymbol{\theta}}, \boldsymbol{z}) = (\boldsymbol{y} - \boldsymbol{x}B\boldsymbol{w})^2$ is the loss function, $M_{\boldsymbol{\theta}_k} = \mathcal{A}_k(S_k)$ represents the model obtained from Empirical Risk Minimization (ERM) algorithm on local dataset $S_k$, i.e., $M_{\boldsymbol{\theta}_k} = \arg \min_M \sum_{i=1}^{n_k} l(M, z_{k,i})$, $M_{\hat{\boldsymbol{\theta}}} = \mathcal{A}(S)$ is the model after one round of FedAvg ($\hat{\boldsymbol{\theta}} = \mathbb{E}_{k \sim \mathcal{K}} \boldsymbol{\theta}_k$), and $\mathcal{K}$ is a uniform distribution overs all nodes.*

*Then, for iid data distribution over nodes, the expected generalization error is*

$$\mathbb{E}_{\{S_k \sim \mathcal{D}_k^{n_k}\}_{k=1}^K} \Delta_{\mathcal{A}}(S) \le \frac{1}{K^2} \sum_{k=1}^K \mathbb{E}_{S_k \sim \mathcal{D}_k^{n_k}} \Delta_{\mathcal{A}_k}(S_k), \tag{18}$$

*and the expected generalization error in non-iid case is*

$$\mathbb{E}_{\{S_k \sim \mathcal{D}_k^{n_k}\}_{k=1}^K} \Delta_{\mathcal{A}}(S) \le \frac{1}{K} \sum_{k=1}^K \mathbb{E}_{S_k \sim \mathcal{D}_k^{n_k}} \Delta_{\mathcal{A}_k}(S_k). \tag{19}$$

*Proof.* Based on Lemma A.1 we obtain

$$\mathbb{E}_{\{S_k \sim \mathcal{D}_k^{n_k}\}_{k=1}^K} \Delta_\mathcal{A}(S) = \mathbb{E}_{k \sim \mathcal{K}, \{S_k, S_k' \sim \mathcal{D}_k^{n_k}\}_{k=1}^K} \left[ \frac{1}{n_k} \sum_{i=1}^{n_k} \left( l(\mathcal{A}(S), z_{k,i}') \right. \right. \tag{20}$$

$$\left. \left. - l(\mathcal{A}(S^{(k,i)}), z_{k,i}') \right) \right]$$

$$= \frac{1}{K} \sum_{k=1}^K \frac{1}{n_k} \sum_{i=1}^{n_k} \mathbb{E}_{\{S_k, S_k' \sim \mathcal{D}_k^{n_k}\}_{k=1}^K} \left[ \left( x_{k,i}' B_{\mathcal{A}(S)} w_{\mathcal{A}(S)} - y_{k,i}' \right)^2 \right. \tag{21}$$

$$\left. - \left( x_{k,i}' B_{\mathcal{A}(S^{(k,i)})} w_{\mathcal{A}(S^{(k,i)})} - y_{k,i}' \right)^2 \right]$$

$$= \frac{1}{K} \sum_{k=1}^K \frac{1}{n_k} \sum_{i=1}^{n_k} \mathbb{E}_{\{S_k, S_k' \sim \mathcal{D}_k^{n_k}\}_{k=1}^K} \left[ \left( x_{k,i}' B_{\mathcal{A}(S)} w_{\mathcal{A}(S)} \right)^2 - \left( x_{k,i}' B_{\mathcal{A}(S^{(k,i)})} w_{\mathcal{A}(S^{(k,i)})} \right)^2 \right. \tag{22}$$

$$\left. - 2 y_{k,i}' \left( x_{k,i}' B_{\mathcal{A}(S)} w_{\mathcal{A}(S)} - x_{k,i}' B_{\mathcal{A}(S^{(k,i)})} w_{\mathcal{A}(S^{(k,i)})} \right) \right]$$

$$= \frac{1}{K} \sum_{k=1}^K \frac{1}{n_k} \sum_{i=1}^{n_k} \mathbb{E}_{\{S_k, S_k' \sim \mathcal{D}^{n_k}\}_{k=1}^K} \left[ - 2 y_{k,i}' x_{k,i}' \left( B_{\mathcal{A}(S)} w_{\mathcal{A}(S)} - B_{\mathcal{A}(S^{(k,i)})} w_{\mathcal{A}(S^{(k,i)})} \right) \right] \tag{23}$$

$$= \frac{1}{K^2} \sum_{k=1}^K \frac{1}{n_k} \sum_{i=1}^{n_k} \mathbb{E}_{\{S_k, S_k' \sim \mathcal{D}^{n_k}\}_{k=1}^K} \left[ - 2 y_{k,i}' x_{k,i}' \left( B_{\mathcal{A}_k(S_k)} w_{\mathcal{A}_k(S_k)} \right. \right. \tag{24}$$

$$\left. \left. - B_{\mathcal{A}_k(S_k^{(i)})} w_{\mathcal{A}_k(S_k^{(i)})} \right) \right]$$

$$= \frac{1}{K^2} \sum_{k=1}^K \frac{1}{n_k} \sum_{i=1}^{n_k} \mathbb{E}_{\{S_k, S_k' \sim \mathcal{D}^{n_k}\}_{k=1}^K} \left[ - 2 y_{k,i}' x_{k,i}' \left( B_{\mathcal{A}_k(S_k)} w_{\mathcal{A}_k(S_k)} \right. \right. \tag{25}$$

$$\left. - B_{\mathcal{A}_k(S_k^{(i)})} w_{\mathcal{A}_k(S_k^{(i)})} \right) + \left( x_{k,i}' B_{\mathcal{A}_k(S_k)} w_{\mathcal{A}_k(S_k)} \right)^2 - \left( x_{k,i}' B_{\mathcal{A}_k(S_k^{(i)})} w_{\mathcal{A}_k(S_k^{(i)})} \right)^2 \Big]$$

$$= \frac{1}{K^2} \sum_{k=1}^K \frac{1}{n_k} \sum_{i=1}^{n_k} \mathbb{E}_{\{S_k, S_k' \sim \mathcal{D}_k^{n_k}\}_{k=1}^K} \left[ \left( x_{k,i}' B_{\mathcal{A}_k(S_k)} w_{\mathcal{A}_k(S_k)} - y_{k,i}' \right)^2 \right. \tag{26}$$

$$\left. - \left( x_{k,i}' B_{\mathcal{A}_k(S_k^{(i)})} w_{\mathcal{A}_k(S_k^{(i)})} - y_{k,i}' \right)^2 \right]$$

$$= \frac{1}{K^2} \sum_{k=1}^K \frac{1}{n_k} \sum_{i=1}^{n_k} \mathbb{E}_{\{S_k, S_k' \sim \mathcal{D}_k^{n_k}\}_{k=1}^K} \left[ l(\mathcal{A}_k(S_k), z_{k,i}') - l(\mathcal{A}_k(S_k^{(i)}), z_{k,i}') \right] \tag{27}$$

$$= \frac{1}{K^2} \sum_{k=1}^K \mathbb{E}_{S_k \sim \mathcal{D}_k^{n_k}} \Delta_{\mathcal{A}_k}(S_k) \tag{28}$$

where (21) is based on uniform $\mathcal{K}$ and squared norm loss. In (23), we assumed iid data distribution over nodes, *i.e.*, $\mathcal{D}_k = ... = \mathcal{D}_K = \mathcal{D}$, so $\mathbb{E}_{\{S_k, S_k' \sim \mathcal{D}_k^{n_k}\}_{k=1}^K} \left( x_{k,i}' B_{\mathcal{A}(S)} w_{\mathcal{A}(S)} \right)^2 = \mathbb{E}_{\{S_k, S_k' \sim \mathcal{D}_k^{n_k}\}_{k=1}^K} \left( x_{k,i}' B_{\mathcal{A}(S^{(k,i)})} w_{\mathcal{A}(S^{(k,i)})} \right)^2$ regardless of $k$. The essential step (24) proceeds by observing that $\mathcal{A}(S^{(k,i)})$ varies solely in the sub-model derived from node $k$, diverging from $\mathcal{A}(S)$, and this discrepancy is magnified by a factor of $K$ when averaging of all sub-models. We can again add appropriate canceled terms in (25) to get the result for iid case. For the general non-iid case, we have

$$\Delta_{\mathcal{A}}(\boldsymbol{S}) = R(\mathcal{A}(\boldsymbol{S})) - R_{\boldsymbol{S}}(\mathcal{A}(\boldsymbol{S})) \tag{29}$$

$$= \frac{1}{K} \sum_{k=1}^{K} \mathbb{E}_{\boldsymbol{z}\sim\mathcal{D}_k} l(\mathcal{A}(\boldsymbol{S}), \boldsymbol{z}) - \frac{1}{K} \sum_{k=1}^{K} \frac{1}{n_k} \sum_{i=1}^{n_k} l(\mathcal{A}(\boldsymbol{S}), \boldsymbol{z}_{k,i}) \tag{30}$$

$$= \frac{1}{K} \sum_{k=1}^{K} \mathbb{E}_{\boldsymbol{z}\sim\mathcal{D}_k} \left( \boldsymbol{x} B_{\mathcal{A}(\boldsymbol{S})} \boldsymbol{w}_{\mathcal{A}(\boldsymbol{S})} - \boldsymbol{y} \right)^2 - \frac{1}{K} \sum_{k=1}^{K} \frac{1}{n_k} \sum_{i=1}^{n_k} l(\mathcal{A}(\boldsymbol{S}), \boldsymbol{z}_{k,i}) \tag{31}$$

$$= \frac{1}{K} \sum_{k=1}^{K} \mathbb{E}_{\boldsymbol{z}\sim\mathcal{D}_k} \left( \frac{1}{K} \sum_{k'=1}^{K} \boldsymbol{x} B_{\mathcal{A}_{k'}(\boldsymbol{S}_{k'})} \boldsymbol{w}_{\mathcal{A}_{k'}(\boldsymbol{S}_{k'})} - \boldsymbol{y} \right)^2 \tag{32}$$

$$- \frac{1}{K} \sum_{k=1}^{K} \frac{1}{n_k} \sum_{i=1}^{n_k} l(\mathcal{A}(\boldsymbol{S}), \boldsymbol{z}_{k,i})$$

$$\leq \frac{1}{K} \sum_{k=1}^{K} \mathbb{E}_{\boldsymbol{z}\sim\mathcal{D}_k} \frac{1}{K} \sum_{k'=1}^{K} \left( \boldsymbol{x} B_{\mathcal{A}_{k'}(\boldsymbol{S}_{k'})} \boldsymbol{w}_{\mathcal{A}_{k'}(\boldsymbol{S}_{k'})} - \boldsymbol{y} \right)^2 \tag{33}$$

$$- \frac{1}{K} \sum_{k=1}^{K} \frac{1}{n_k} \sum_{i=1}^{n_k} l(\mathcal{A}(\boldsymbol{S}), \boldsymbol{z}_{k,i})$$

$$\leq \frac{1}{K} \sum_{k'=1}^{K} \left( \frac{1}{K} \sum_{k=1}^{K} \mathbb{E}_{\boldsymbol{z}\sim\mathcal{D}_k} \left( \boldsymbol{x} B_{\mathcal{A}_{k'}(\boldsymbol{S}_{k'})} \boldsymbol{w}_{\mathcal{A}_{k'}(\boldsymbol{S}_{k'})} - \boldsymbol{y} \right)^2 \right) \tag{34}$$

$$- \frac{1}{K} \sum_{k=1}^{K} \min_{M} \frac{1}{n_k} \sum_{i=1}^{n_k} l(M, \boldsymbol{z}_{k,i})$$

$$= \frac{1}{K} \sum_{k'=1}^{K} R(\mathcal{A}_{k'}(\boldsymbol{S}_{k'})) - \frac{1}{K} \sum_{k=1}^{K} R_{\boldsymbol{S}_k}(\mathcal{A}_k(\boldsymbol{S}_k)) \tag{35}$$

$$= \frac{1}{K} \sum_{k=1}^{K} \Delta_{\mathcal{A}_k}(\boldsymbol{S}_k), \tag{36}$$

where (32) is based on averaging in FedAvg. (33) is derived following the convexity of squared norm.

$\square$

## B  PROOF OF THEOREM 4.2

**Theorem B.1.** *Consider that $M_{\boldsymbol{\theta}}(\boldsymbol{x}) = \boldsymbol{x} B \boldsymbol{w}$ is the model class, $l(M_{\boldsymbol{\theta}}, \boldsymbol{z}) = (\boldsymbol{y} - \boldsymbol{x} B \boldsymbol{w})^2$ is the loss function, and $\mathcal{K}$ is a uniform distribution over all nodes. Local models at round $r$ are calculated by doing multiple local steps as in (6). The aggregated model at round $r$ is $M_{\hat{\boldsymbol{\theta}}_r}$ is obtained by performing FedAvg in (5) where the data points used in round $r$ (i.e., $Z_{k,r}$) are sampled without replacement.*

*For iid data distribution over nodes, the average generalization error bound is expressed as*

$$\mathbb{E}_{\{\boldsymbol{S}_k\sim\mathcal{D}_k^{n_k}\}_{k=1}^K} \Delta_{FedAvg}(\boldsymbol{S}) \leq \frac{1}{R} \sum_{r=1}^{R} \frac{1}{K^2} \sum_{k=1}^{K} \sqrt{\frac{\mathcal{C}(M_{\boldsymbol{\theta}})}{|Z_{k,r}|}}, \tag{37}$$

*while in non-iid case, the average generalization error bound is expressed as*

$$\mathbb{E}_{\{\boldsymbol{S}_k\sim\mathcal{D}_k^{n_k}\}_{k=1}^K} \Delta_{FedAvg}(\boldsymbol{S}) \leq \frac{1}{R} \sum_{r=1}^{R} \frac{1}{K} \sum_{k=1}^{K} \sqrt{\frac{\mathcal{C}(M_{\boldsymbol{\theta}})}{|Z_{k,r}|}}, \tag{38}$$

*where $\mathcal{C}(M_{\boldsymbol{\theta}})$ shows the complexity of the model $M_{\boldsymbol{\theta}}$.*

---

**Algorithm 2** FedAvg

---

**Input**: Initial model $\{\boldsymbol{\theta}_{k,1,0}\}_{k=1}^{K}$, Learning rate $\eta$, and number of local steps $\tau$.

**Output**: $\hat{\boldsymbol{\theta}}_R$

1: **for** Round $r$ in $1, ..., R$ **do**
2:    **for** Node $k$ in $1, ..., K$ **in parallel do**
3:       **for** Local step $t$ in $0, ..., \tau - 1$ **do**
4:          Sample the batch $\mathcal{B}_{k,r,t}$ from $\mathcal{D}_k$.
5:          $\boldsymbol{\theta}_{k,r,t+1} = \boldsymbol{\theta}_{k,r,t} - \frac{\eta}{|\mathcal{B}_{k,r,t}|} \sum_{i \in \mathcal{B}_{k,r,t}} \nabla l(M_{\boldsymbol{\theta}_{k,r,t}}, \boldsymbol{z}_{k,i})$
6:       $\boldsymbol{\theta}_{k,r+1,0} = \frac{1}{K} \sum_{k=1}^{K} \boldsymbol{\theta}_{k,r,\tau}$
7: **return** $\hat{\boldsymbol{\theta}}_R = \frac{1}{K} \sum_{k=1}^{K} \boldsymbol{\theta}_{k,R,\tau}$

---

*Proof.* For non-iid case, we have

$$\mathbb{E}_{\{\boldsymbol{S}_k \sim \mathcal{D}_k^{n_k}\}_{k=1}^{K}} \Delta_{FedAvg}(\boldsymbol{S}) = \mathbb{E}\{\boldsymbol{S}_k \sim \mathcal{D}_k^{n_k}\}_{k=1}^{K} \frac{1}{KR} \sum_{r=1}^{R} \sum_{k=1}^{K} \Bigg( \mathbb{E}_{\boldsymbol{z} \sim \mathcal{D}_k} l(M_{\hat{\boldsymbol{\theta}}_r}, \boldsymbol{z}) \tag{39}$$

$$- \frac{1}{|Z_{k,r}|} \sum_{i \in Z_{k,r}}^{K} l(M_{\hat{\boldsymbol{\theta}}_r}, \boldsymbol{z}_{k,i}) \Bigg)$$

$$= \frac{1}{KR} \sum_{r=1}^{R} \sum_{k=1}^{K} \mathbb{E}\{Z_{k,r} \sim \mathcal{D}_k^{|Z_{k,r}|}\}_{k=1}^{K} \Bigg( \mathbb{E}_{\boldsymbol{z} \sim \mathcal{D}_k} l(M_{\hat{\boldsymbol{\theta}}_r}, \boldsymbol{z}) \tag{40}$$

$$- \frac{1}{|Z_{k,r}|} \sum_{i \in Z_{k,r}}^{K} l(M_{\hat{\boldsymbol{\theta}}_r}, \boldsymbol{z}_{k,i}) \Bigg)$$

$$= \frac{1}{R} \sum_{r=1}^{R} \mathbb{E}\{Z_{k,r} \sim \mathcal{D}_k^{|Z_{k,r}|}\}_{k=1}^{K} \Delta_{\mathcal{A}}(\{Z_{k,r}\}_{k=1}^{K}) \tag{41}$$

$$\leq \frac{1}{R} \sum_{r=1}^{R} \frac{1}{K} \sum_{k=1}^{K} \mathbb{E}\{Z_{k,r} \sim \mathcal{D}_k^{|Z_{k,r}|}\} \Delta_{\mathcal{A}_k}(Z_{k,r}) \tag{42}$$

$$\leq \frac{1}{R} \sum_{r=1}^{R} \frac{1}{K} \sum_{k=1}^{K} \sqrt{\frac{\mathcal{C}(M_{\boldsymbol{\theta}})}{|Z_{k,r}|}}, \tag{43}$$

where in (41), $\mathcal{A}$ represents one-round FedAvg algorithm. In (42) we have used lemma 4.1. The same proof applies to the iid case. □

## C   ALGORITHMS USED IN THE EXPERIMENTS

In this section we have listed our implementation of FedAvg (Algorithm 2), SCAFFOLD Karimireddy et al. (2019) (Algorithm 3), and FedALS+SCAFFOLD (Algorithm 4). In Algorithm 3, we observe that in addition to the model, SCAFFOLD also keeps track of a state specific to each client, referred to as the client control variate $\boldsymbol{c}_{k,r}$. It is important to recognize that the clients within SCAFFOLD have memory and preserve the $\boldsymbol{c}_{k,r}$ and $\sum_{k=1}^{K} \boldsymbol{c}_{k,r}$ values. Additionally, when $\boldsymbol{c}_{k,r}$ consistently remains at $0$, SCAFFOLD essentially becomes equivalent to FedAvg.

Algorithm 4 demonstrate the integration of FedALS and SCAFFOLD. It is important to observe that in this algorithm, the control variables are fragmented according to various model layers, and there exist distinct local step counts for different layers. This concept underlies the fundamental principle of FedALS.

---

**Algorithm 3** SCAFFOLD

---

**Input**: Initial model $\{\boldsymbol{\theta}_{k,1,0}\}_{k=1}^{K}$, Initial control variable $\{\boldsymbol{c}_{k,1}\}_{k=1}^{K}$, learning rate $\eta$, and number of local steps $\tau$.

**Output**: $\hat{\boldsymbol{\theta}}_R$

1: **for** Round $r$ in $1, ..., R$ **do**
2:  **for** Node $k$ in $1, ..., K$ **in parallel do**
3:    **for** Local step $t$ in $0, ..., \tau - 1$ **do**
4:      Sample the batch $\mathcal{B}_{k,r,t}$ from $\mathcal{D}_k$.
5:        $\boldsymbol{\theta}_{k,r,t+1} = \boldsymbol{\theta}_{k,r,t} - \eta\big(\frac{1}{|\mathcal{B}_{k,r,t}|}\sum_{i \in \mathcal{B}_{k,r,t}}\nabla l(M_{\boldsymbol{\theta}_{k,r,t}}, \boldsymbol{z}_{k,i}) - \boldsymbol{c}_{k,r} + \sum_{k=1}^{K}\boldsymbol{c}_{k,r}\big)$
6:      $\boldsymbol{c}_{k,r+1} = \sum_{k=1}^{K}\boldsymbol{c}_{k,r} - \boldsymbol{c}_{k,r} + \frac{1}{\eta\tau}(\boldsymbol{\theta}_{k,r,0} - \boldsymbol{\theta}_{k,r,\tau})$
7:      $\boldsymbol{\theta}_{k,r+1,0} = \frac{1}{K}\sum_{k=1}^{K}\boldsymbol{\theta}_{k,r,\tau}$
8: **return** $\hat{\boldsymbol{\theta}}_R = \frac{1}{K}\sum_{k=1}^{K}\boldsymbol{\theta}_{k,R,\tau}$

---

**Algorithm 4** FedALS + SCAFFOLD

---

**Input**: Initial model $\{\boldsymbol{\theta}_{k,1,0} = [\boldsymbol{\phi}_{k,1,0}, \boldsymbol{h}_{k,1,0}]\}_{k=1}^{K}$, Initial control variable $\{\boldsymbol{c}_{k,1} = [\boldsymbol{c}_{k,1}^{\boldsymbol{\phi}}, \boldsymbol{c}_{k,1}^{\boldsymbol{h}}]\}_{k=1}^{K}$, learning rate $\eta$, number of local steps for the head model $\tau$, adaptation coefficient $\alpha$.

**Output**: $\hat{\boldsymbol{\theta}}_R$

1: **for** Round $r$ in $1, ..., R$ **do**
2:  **for** Node $k$ in $1, ..., K$ **in parallel do**
3:    **for** Local step $t$ in $0, ..., \tau - 1$ **do**
4:      Sample the batch $\mathcal{B}_{k,r,t}$ from $\mathcal{D}_k$.
5:        $\boldsymbol{\theta}_{k,r,t+1} = \boldsymbol{\theta}_{k,r,t} - \eta\big(\frac{1}{|\mathcal{B}_{k,r,t}|}\sum_{i \in \mathcal{B}_{k,r,t}}\nabla l(M_{\boldsymbol{\theta}_{k,r,t}}, \boldsymbol{z}_{k,i}) - \boldsymbol{c}_{k,r} + \sum_{k=1}^{K}\boldsymbol{c}_{k,r}\big)$
6:      **if** $\mod{(r\tau + t, \tau)} = 0$ **then**
7:        $\boldsymbol{c}_{k,r}^{\boldsymbol{h}} \leftarrow \frac{1}{K}\sum_{k=1}^{K}\boldsymbol{c}_{k,r}^{\boldsymbol{h}} - \boldsymbol{c}_{k,r}^{\boldsymbol{h}} + \frac{1}{\eta\tau}(\boldsymbol{h}_{k,r,0} - \boldsymbol{h}_{k,r,t})$
8:        $\boldsymbol{h}_{k,r,t} \leftarrow \frac{1}{K}\sum_{k=1}^{K}\boldsymbol{h}_{k,r,t}$
9:      **else if** $\mod{(r\tau + t, \alpha\tau)} = 0$ **then**
10:        $\boldsymbol{c}_{k,r}^{\boldsymbol{\phi}} \leftarrow \frac{1}{K}\sum_{k=1}^{K}\boldsymbol{c}_{k,r}^{\boldsymbol{\phi}} - \boldsymbol{c}_{k,r}^{\boldsymbol{\phi}} + \frac{1}{\eta\alpha\tau}(\boldsymbol{\phi}_{k, \lfloor\frac{r\tau+t-\alpha\tau}{\tau}\rfloor,\ \mod{(r\tau+t-\alpha\tau, \tau)}}^{l} - \boldsymbol{\phi}_{k,r,t}^{l})$
11:        $\boldsymbol{\phi}_{k,r,t} \leftarrow \frac{1}{K}\sum_{k=1}^{K}\boldsymbol{\phi}_{k,r,t}$
12:    $\boldsymbol{c}_{k,r+1} = \boldsymbol{c}_{k,r}$
13:    $\boldsymbol{\theta}_{k,r+1,0} = \boldsymbol{\theta}_{k,r,\tau}$
14: **return** $\hat{\boldsymbol{\theta}}_R = \frac{1}{K}\sum_{k=1}^{K}\boldsymbol{\theta}_{k,R,\tau}$

---

