# OpenReview forum: "Communication Efficient Federated Representation Learning"
_ICLR.cc/2024/Conference — ICLR 2024 Conference Withdrawn Submission_

### Official Review · Reviewer_1qmZ · 2023-10-30

**Soundness:** 3 good
**Presentation:** 2 fair
**Contribution:** 3 good
**Rating:** 3
**Confidence:** 3

**Summary:**

The paper delves into the realm of federated learning, specifically focusing on the concept of representation learning to enhance the efficiency of federated learning systems. The authors introduce a novel approach, termed "FedALS," which aims to optimize the learning process by increasing the number of local steps for the representation extractor. This optimization is shown to enhance performance across various setups, irrespective of the number of clients, and even in scenarios with heterogeneous (non-iid) data distributions.

A significant highlight of the paper is its emphasis on the importance of generalization in machine learning systems. The authors draw upon statistical learning theory to discuss the generalization error and its implications in federated learning. They reference prior work by Barnes et al. (2022a) that provided upper bounds on the expected generalization error for federated learning in an iid setting. Building on this, the authors present their results for both iid and non-iid settings, specifically for the linear representation model class.

The experimental section of the paper evaluates the performance of FedALS using ResNet-20 as a deep neural network architecture. The experiments, conducted on datasets like CIFAR-10, CIFAR-100, and SVHN, demonstrate the anticipated performance improvements of the proposed approach, especially in non-iid settings.

**Strengths:**

1.  The paper introduces "FedALS," a novel method that optimizes federated learning by increasing the number of local steps for the representation extractor. This unique approach demonstrates enhanced performance across various setups, showcasing its adaptability and effectiveness in diverse federated learning scenarios.
2.  The authors place significant emphasis on the concept of generalization in machine learning, a critical aspect that determines the efficacy of any learning system. By leveraging statistical learning theory and discussing the generalization error, the paper provides valuable insights into the challenges and potential solutions for achieving better generalization in federated learning contexts.

**Weaknesses:**

The paper might lack an extensive comparative analysis with existing state-of-the-art federated learning methods. While the introduction of FedALS is promising, understanding its performance in relation to well-established techniques is crucial. A detailed comparative study, encompassing a variety of performance metrics, datasets, and neural network architectures, would provide clearer insight into the method's relative advantages and potential limitations. For example, the authors can extend the experiments into the NLP field too. ResNet-20 is also a too-old NN architecture.

**Questions:**

1. What are the long-term implications of using FedALS, especially in large-scale, real-world deployments? How does the method handle challenges like model drift, evolving data distributions, or scalability constraints in extensive federated networks? Are there any inherent limitations or challenges in the FedALS approach that might need to be addressed for its broader adoption in practical applications?
2. How does FedALS perform in more complex or diverse federated environments, especially those with unique constraints or challenges? Can the authors provide insights or additional experiments that demonstrate the method's adaptability and robustness in such varied contexts?

---

### Official Review · Reviewer_iXzj · 2023-11-03

**Soundness:** 3 good
**Presentation:** 3 good
**Contribution:** 3 good
**Rating:** 5
**Confidence:** 4

**Summary:**

This paper proposed a new algorithm for federated learning where the feature extractors are aggregated less frequently compared to the classifier head. Empirical experiments on SVHN, CIFAR-10, CIFAR-100 suggest the proposed algorithm also improves the generalization accuracy on test data.

==== after rebuttal =====

No response in the discussion phase. Keep the borderline rejection score.

**Strengths:**

The proposed method is simple and achieves good empirical results.

**Weaknesses:**

Maybe I missed it, the theory contribution in section 4 is not clear to me. Could the authors clarify (1) the connection to the later algorithm that separates feature extractor and classifier head? and (2)
the relationship to Collins et al. 2021;2022?

For the empirical results, could the authors provide more details on hyperparameters and how they are tuned?

What is the aggregation frequency of FedAvg? It might be good to compare with both \tau=5, and \tau*\alpha=50, and somewhere in between like 20.

It would also be nice to run experiments on a dataset with natural heterogeneity, e.g., client partition based on users.  Or at least discuss the potential application scenario, e.g., following ​​https://arxiv.org/abs/2107.06917 Section 3.1. The “artificial” heterogeneity of 2 classes per client may result in different behavior as suggested by https://arxiv.org/abs/2206.04723.

**Questions:**

Could the authors provide more details about the complexity definition in Eq (11) (12)?

---

### Official Review · Reviewer_w69N · 2023-11-04

**Soundness:** 1 poor
**Presentation:** 2 fair
**Contribution:** 1 poor
**Rating:** 3
**Confidence:** 3

**Summary:**

This paper has shown that for data that is not identically distributed among nodes in FL, errors tend to be less in the initial layers of a neural network compared to the later ones. This suggests that initial layers, which help the network understand input data, might not need to be averaged as often. Further analysis, especially focusing on linear representation, reveals that less frequent averaging or more local training before averaging can make the final model more accurate. Experimental results partly support the effectiveness of this approach.

**Strengths:**

- This paper explores the deviation degree between clients according to the depth of the deep neural network. This study reveals that early layers diverge less since these layers focus on learning representations of inputs.

- Experiments demonstrate that the proposed method works well for federated learning (FL) in situations where data is either identically or non-identically distributed across different nodes.

**Weaknesses:**

- the proposed method seems to be unstable. The oscillation during the training phase seems much larger than baselines (FedAvg, SCAFFOLD) and its performance to highly dependent on the selection of $\alpha$ which is a hyper-parameter.

- The suggested method needs all participants to be involved in the training throughout the learning process, which is not feasible in Federated Learning due to its distributed nature.

- More extensive experiment is required
  - A thorough comparison with state-of-the-art (SOTA) methods is essential to validate the proposed method. Given its applicability to the standard Federated Learning (FL), it is crucial to benchmark its performance against SOTA methods such as FedDC, FedMLB, FedDyn, and MOON. If the method can be used with these established techniques, it's crucial to demonstrate that combining the proposed method with them leads to significant benefits.

  - To validate its effectiveness on heterogeneous data, the method needs evaluation across a broader range of scenarios with different levels of data diversity.

- Paper claims that the proposed method reduces communication costs in Federated Learning when training a global model, but it lacks numerical data or theoretical evidence to back up this claim.

**Questions:**

Please refer to the Weaknesses Section.

---

### Official Review · Reviewer_fepL · 2023-11-10

**Soundness:** 2 fair
**Presentation:** 4 excellent
**Contribution:** 2 fair
**Rating:** 3
**Confidence:** 4

**Summary:**

This paper considers FedAvg where a set of clients collaboratively learn a common model in a distributed setting. They make the observation that :  the consensus error in the initial layers of a deep neural network is considerably smaller than that observed in the final layers => applying a less intensive averaging approach for the initial layers. Motivated by this they propose increasing
the number of local SGD iterations or reducing the frequency of averaging for the representation extractor (lower layers aka encoder) leads to enhanced generalizability in the learned model.

**Strengths:**

* The main contribution of the paper is to do update the rep layers less frequently and the linear head more frequently resulting in reduced communication bottleneck -- Overall I find the idea quite intuitive and clever.

**Weaknesses:**

* Overall, while the idea is cute --- I don't feel that the claims are obvious from the theory / experiments. Further, Given https://proceedings.mlr.press/v139/collins21a/collins21a.pdf , the current paper seems slightly incremental.

* While the intuition makes sense, Theorem 2 does not say anything about benefits of treating encoder and linear head separately. It simply says smaller the models (low complexity) tighter the UB. I feel the theoretical motivation from Theorem 2 is slightly unclear to me.
Also, this theorem is not too interesting to me actually ~ this is kind of obvious, no ?

* Further the theoretical motivation to set $\tau_{\phi} \propto \frac{C(M_\phi)}{C(M_h)}\tau_h$ does not say anything about the earlier vs later layers .... *my interpretation* is that it simply implies if I choose two separate blocks of the model to be updated separately (i.e. at different frequency) we should update the larger block less frequently.
While the attempt to connect this with the nice representation learning interpretation is nice ~~ it is not clear if this is obvious from this result.

* *I would encourage the authors to conduct the following simple experiment to verify : Take a N layer model ~ and run the experiment with different block combinations for $h$ and $\phi$ i.e. 1-(N-1), 2-(N-2), ... , (N-1)-1 layers  ....*

* From Fig 3 : I suspect that FedALS results in more variance in the non iid setting? In iid setting for both SVHN and CIFAR10 FedALS has lower accuracy and weaker convergence properties ? *I also encourage the authors to repeat the experiment with different $\tau$*

*  Also I encourage the authors to report the confidence intervals in Table 1 for both iid and non -iid setting. Also for Fig 3 please plot the variances for iid also ?

**Questions:**

See Weakness
+ i am open to revisiting my evaluation if you can convincingly address my issues during the discussion phase.